# Promotion of Ru or Ni on Alumina Catalysts with a Basic Metal for CO_2_ Hydrogenation: Effect of the Type of Metal (Na, K, Ba)

**DOI:** 10.3390/nano12071052

**Published:** 2022-03-23

**Authors:** Enrique García-Bordejé, Ana Belén Dongil, José M. Conesa, Antonio Guerrero-Ruiz, Inmaculada Rodríguez-Ramos

**Affiliations:** 1Department of Chemical Processes and Nanotechnology, Instituto de Carboquímica (ICB-CSIC), Miguel Luesma Castán 4, 50018 Zaragoza, Spain; 2Institute of Catalysis and Petrochemistry, CSIC, 28049 Madrid, Spain; a.dongil@csic.es (A.B.D.); jmconesa@ccia.uned.es (J.M.C.); irodriguez@icp.csic.es (I.R.-R.); 3Departamento de Química Inorgánica y Química Técnica, UNED, Av. de Esparta s/n, 28232 Las Rozas de Madrid, Spain; aguerrero@ccia.uned.es; 4Grupo de Diseño y Aplicación de Catalizadores Heterogéneos, Unidad Asociada UNED-CSIC (ICP), 28049 Madrid, Spain

**Keywords:** CO_2_ methanationt, base metal, catalyst promotion, Ru, Ni

## Abstract

Ru and Ni on alumina catalysts have been promoted with a 10 wt% of alkali metal (K or Na) or alkaline earth metal (Ba) and tested in CO_2_ methanation. For the catalyst consisting of Ni and Ba, the variation of Ba loading while keeping Ni loading constant was studied. The promotion in terms of enhanced CH_4_ yield was found only for the addition of barium to 15 wt% Ni/Al_2_O_3_. In contrast, K and Na addition increased the selectivity to CO while decreasing conversion. For the Ru-based catalyst series, no enhancement in conversion or CH_4_ yield was attained by any of the alkaline metals. CO_2_ temperature-programed desorption (CO_2_-TPD) revealed that the amount of chemisorbed CO_2_ increased significantly after the addition of the base metal. The reactivity of CO_x_ ad-species for each catalyst was assessed by temperature-programed surface reaction (TPSR). The characterization revealed that the performance in the Sabatier reaction was a result of the interplay between the amount of chemisorbed CO_2_ and the reactivity of the COx ad-species, which was maximized for the (10%Ba)15%Ni/Al_2_O_3_ catalyst.

## 1. Introduction

Urgent measures are needed to stop global warming due to anthropogenic emissions [1]. In this scenario, the Green Deal of the European Commission proposed in September 2020 to raise the 2030 greenhouse gas emission reduction target to at least 55% compared to 1990. Among the key targets of the EU for 2030 are at least 40% cuts in greenhouse gas emissions (from 1990 levels) and at least 32% share for renewable energy [2]. The main hurdles of renewable energy are its intermittency and misalignment with the demand. Therefore, there is an urgent need to find a suitable method to store renewable energy. Renewable energy can be converted into H_2_ by water electrolysis, which is a mature technology. The main drawback of H_2_ is the difficulty of its storage and transport, but it can be converted to CH_4_ by reaction with captured CO_2_. CH_4_ can be advantageously stored using the current natural gas infrastructures. Thus, the conversion of H_2_ and captured CO_2_ via the Sabatier reaction meets the two objectives of the European Green Deal, i.e., being an enabling technology to increase the share of renewable energy and greenhouse gas reduction, contributing to carbon capture and utilization (CCU).

There have been extensive research efforts trying to achieve high catalytic activity for CO_2_ methanation, especially in the low-temperature region (100–250 °C). The state-of-the-art catalysts are based on Ni and Ru supported on a high-surface metal oxide-containing basic sites. The Ni and Ru metal perform the dissociative chemisorption of H_2_ while the basic sites of the support activate the mildly acidic CO_2_ molecule [3]. The CO_2_ ad-species (such as carbonates) undergo hydrogenation by reacting with a hydrogen spilt over from the metal nanoparticles. It has been reported that the basic sites responsible for CO_2_ activation are those which adsorb CO_2_ with intermediate strength [4,5,6,7]. Neither CO_2_ ad-species that are bound too weakly nor carbonates being too stable in the temperature range of methanation would probably participate in the reaction. Basic supports with different Lewis (lattice oxygen and oxygen vacancies) and Brönsted basicity (hydroxyl groups) have been screened for methanation reaction [8,9]. It was observed that the activity of the catalysts depended on the reaction temperature, and the best turnover was attained using supports exhibiting a medium surface basicity. Another way to increase catalyst basicity is by adding alkaline and alkaline-earth metals [10]. Additionally, the alkaline promoter is also electropositive. Therefore, it may also decrease the work function of the methanation metal, i.e., increasing the Lewis basicity of the metal [11]. Falconer et al. [12] found that K addition to Ni/Al_2_O_3_ increased the CO_2_ methanation kinetics for low K loadings and changed the CH_4_/CO distribution. A large number of papers study the effect of the addition of one single metal selected among the different metals of group I (mainly Na or K) and group II (mainly Ca or Mg). According to the revised literature, conflicting results are reported for the same alkali and alkali-earth addition to methanation catalyst, sometimes reporting promotion of the CO_2_ methanation activity while others reporting suppression of it. This behavior variability stems from the fact that the performance is very sensitive to the preparation method, alkaline loadings and reaction conditions. Some works [13,14] report that the CO_2_ conversion and CH_4_ selectivity decreased when adding the alkaline with respect to the monometallic catalyst [13]. It was found that the performance of the monometallic catalyst in CO_2_ hydrogenation worsened when the alkaline metal was added. The explanation usually given is that the alkaline can mask Ru (or Ni) or that stable alkaline carbonates can be formed. It is generally reported that the selectivity to CO increases when adding even small amounts of Na [15] or K [16,17]. At odds with the above-mentioned works, increased CO_2_ conversion and selectivity to CH_4_ are also reported for low K loadings [18]. In other work [19] a positive promotion of the Ni/SO_2_ catalyst for CH_4_ production by in-situ doping with K (0.6 wt%) from KOH in aerosols coming from a previous electrolyzer was found. The promotion was not occurring when a catalyst with the same K content was prepared by KOH wet impregnation. Improved CO_2_ methanation was also reported for Mg addition to Ni-based catalyst on different supports such as carbon support [20], MCM-41 [21] and zeolites [22,23].

In works that study the effect of the promoter loading, the promotion by the alkaline usually exhibits a volcano-type behavior as a function of the alkaline loading [24,25]. The promotion was found only for smaller Ru nanoparticles (low loading) and low alkaline loadings [24]. The promoting effect is maximized for certain loading, and further loading masked the catalyst, decreasing activity and selectivity [26]. Takano et al. found an optimum Ca concentration (17–20% Ca) by impregnation of Ni and Ca on ZrO_2_ [27]. An optimum Mg loading is also found for Ni on different supports [25,26,28]. The claimed benefits are not only the increased basicity and CO_2_ chemisorption [25,29,30] but also the enhancement of Ni dispersion and resistance to sintering [28] and the creation of oxygen vacancies on the support [31,32].

Since the type of alkaline also affects the performance, some authors screened several alkaline or alkaline-earth metals added to Ru/Al_2_O_3_ or Ni/Al_2_O_3_ catalysts and compared their catalytic performance [14,24,33,34]. Cimino et al. [34] impregnated different alkaline metals (Li, Na, K) over 1%Ru/Al_2_O_3_ and found a promotion only when Li was impregnated. Liang et al. [35] added up to 16 metal additives, including alkali and alkaline earth metals, to 1%Ni/Al_2_O_3_. None of these catalysts provided enhanced performance due to the strong CO_2_ absorption as carbonates. Although lighter alkaline earths (Mg, Ca) have been extensively studied, little research has been conducted on the promotion of heavier alkaline earth metals such as Sr or Ba, despite the fact that the previous screening of those metals reported enhanced performance [36,37,38]. Sr or Ba enhanced methanation while Mg or Ca addition promoted RWGS reaction [36]. The activity increased significantly upon Sr and Ba addition, especially in the low-temperature region [36,38]. This is in conflict with the work of Büchel et al. [17], who do not report any enhanced activity as a result of the addition of K or Ba. Thus, further research is needed about the promotion with heavier alkaline earth metals such as barium and about their comparison with the promotion by alkali metals. 

We previously studied the positive effect in CO_2_ methanation of increasing the basicity of carbon nanofibers by nitrogen doping [39] or by the formation of bimetallic NiRu particles [7]. Here, we added different alkali metals (K, Na) or an alkaline earth metal (Ba) as basic promoters to Ru and Ni on alumina catalyst and compared the effect on methanation reaction. Some preparation parameters investigated are the order of metal impregnation (promoter and noble metal) and the amount of the promoter. The amount of chemisorbed CO_2_ in each catalyst was characterized by temperature-programed desorption of pre-adsorbed CO_2_ (CO_2_-TPD) and the reactivity of adsorbed CO_2_ by temperature-programed surface reaction (TPSR). This allowed the disclosure of some differences in the behavior between the alkali or alkali earth promoters and the two different methanation metals.

## 2. Experimental

### 2.1. Catalyst Preparation

Alumina support was prepared from Pural (SASOL) calcined at 500 °C in air. The chemical precursors for the metals were Ni(NO_3_)_2_·6H_2_O (Sigma-Aldrich) and Ru(NO_3_)_3_NO (Alfa Aesar). As a precursor of the alkaline metals, K_2_CO_3_, NaNO_3_ and Ba(NO_3_)_2_ (Sigma-Aldrich) were used. 

Due to the different costs of the metals in the current market, i.e., 9550 €/Kg and 18.34 €/Kg for Ru and Ni, respectively, the nickel loading was set at 15 wt% while the ruthenium loading was set at a considerably lower value (3 wt%). The higher loading of nickel also influenced the decision about the order of impregnation of the transition and alkaline metal. For the nickel catalyst, the nickel was impregnated first and subsequently the alkaline metal, while for Ru, the order of impregnation was the reverse, i.e., first the alkaline and subsequently Ru to prevent the Ru from being buried by the alkaline. For comparison, the reverse order of addition was also studied for the Ru-containing catalyst. 

In the case of the Ni catalyst series, the amount of precursor was weighted to set the Ni loading on γ-alumina at 15 wt%, diluted in the suitable amount of water to impregnate γ-alumina by incipient wetness impregnation. After drying at 110 °C, the catalyst was calcined at 500 °C under N_2_ for 1 h and subsequently reduced in H_2_ at the same temperature for 1 h. Subsequently, the alkaline precursor was impregnated to yield a final loading of 10 wt% of metal with respect to alumina. After drying at 110 °C, the material was calcined at 500 °C under N_2_ for 1 h.

In the case of the Ru catalyst series, the standard preparation method was the impregnation of the alkaline metal first, and later, the methanation metal for the reason explained above. The alkaline metal precursor was weighted to yield a final loading of 10 wt% of the metal element with respect to γ-alumina weight, diluted in water and impregnated on alumina by incipient wetness impregnation. After drying at 110 °C, the material was calcined at 500 °C under N_2_. Subsequently, the amount of Ru precursor was weighted to the set 3 wt% with respect to alumina. After drying at 110 °C, the material was calcined at 500 °C under N_2_ flow for 1 h and subsequently reduced under H_2_ flow at the same temperature for 1h. For comparison, the reverse order of addition was also prepared. The samples are denoted using the following nomenclature: “(second impregnated metal) first impregnated metal/Al”. The actual metal loadings were measured by ICP-OES, and the results were within ±5% of the nominal loading.

### 2.2. Catalytic Testing

Catalytic testing was carried out in a continuous-flow 6 mm-outer-diameter quartz reactor inside a vertical furnace equipped with a temperature controller (Eurotherm). The amount of monometallic catalyst used in the catalytic tests was 50 mg. For the bimetallic catalyst, the amount of catalyst was adjusted to have the same Ru or Ni weight as the monometallic catalyst. Subsequently, the catalyst was diluted with SiC and placed inside the reactor forming a packed bed with a thermocouple inside the bed. Prior to catalytic tests, the catalyst was heated to 500 °C in a N_2_ flow using a heating rate of 10 °C min^−1^, and it was reduced with H_2_ (60 mL min^−1^) at 500 °C for 1 h. The reaction temperature was controlled with a thermocouple inside the catalytic bed. The reaction conversion and selectivities were recorded at steady-state using a 60 mL min^−1^ reaction mixture consisting of 5 % CO_2_, 20% H_2_ and Ar as balance. This flow rate provided a gas hourly space velocity (GHSV) of 19,000 h^−1^. Gas analysis was performed using a Pfeiffer vacuum mass spectrometer. The following *m*/*z* signals were recorded in a mass spectrometer: 2, 16, 18, 28, 40, 44. The signals of the gases were calibrated considering the baseline of Ar and the fragmentation pattern of each mass. The main *m*/*z* signals used for each gas were 2 (H_2_), 16 (CH_4_), 18 (H_2_O), 28 (CO), 40 (Ar) and 44 (CO_2_). The concentration of CO was calculated by subtracting the contribution of CO_2_ from *m*/*z* = 28. The concentration of CH_4_ was calculated by subtracting the contribution of CO_2_, CO and H_2_O from *m*/*z* = 16. The calibration was carried out with gas cylinders of known concentrations of each gas. The correct calibration of the mass spectrometer was double-checked, analyzing the gases using a calibrated Agilent Micro GC 3000 A. 

For the calculation of the kinetic parameters, first, the Weisz–Prater criterion was applied to assess the absence of internal and external mass transfer limitations [7]. The kinetic rate constants (*k*) were calculated considering a differential reactor and isothermal conditions, guaranteed by the small reactor diameter, bed length and dilution with SiC. To calculate the reaction rate, first, we used the equation of a differential reactor [40]; the reaction is assumed to be a zero-order reaction with respect to the CO_2_ concentration as observed for similar catalysts [41], whereby CO_2_ reacts from the adsorbed state.

To calculate the apparent activation energy, the linearization of the Arrhenius equation was applied (Equation (1)):(1)$$\ln k=\ln A-\frac{E_{a}}{RT}$$
where *k* is the kinetic constant in mmol CO_2_ min^−1^ mg^−1^, *A* is an exponential factor and *E*_a_ is the apparent activation energy, *T* is the temperature in Kelvin and *R* is the ideal gas constant, i.e., 8.314 J mol^−1^ K^−1^. To calculate the apparent activation energy, it is necessary to identify the temperature range of the kinetic regime, i.e., whereby the plot of ln *k* vs. 1/*T* is linear. This occurs for low temperatures; that is, before entering the diffusion-limited regime.

### 2.3. Catalyst Characterization

The catalysts were characterized by transient techniques, namely, temperature-programed desorption of pre-adsorbed CO_2_ (CO_2_-TPD) and temperature-programed surface reaction (TPSR). These experiments were conducted in the same setup as catalytic testing. The purpose of CO_2_-TPD experiments is to quantify the CO_2_ chemisorbed at 300 °C. To this end, the catalyst was heated to 500 °C at a heating rate of 10 K min^−1^ in inert gas. At this temperature, the catalyst was reduced with 100 mL min^−1^ of H_2_ mixture for 1 h. Subsequently, the temperature was set at 300 °C and 60 mL min^−1^ of CO_2_ was flushed for 1 h. The gas was switched to 60 mL min^−1^ Ar and the reactor was allowed to cool down to room temperature. The Ar flow was kept constant overnight to remove all weakly physisorbed CO_2_. Then the gas was adjusted to 60 mL min^−1^ of Ar and, when the signal of the mass spectrometer was stable, the temperature was increased to 500 °C at a rate of 10 °C per minute while monitoring the desorbed gases, mainly CO_2_.

The main purpose of the TPSR experiments is to determine the temperature at which CH_4_ starts to evolve from the reaction of H_2_ gas with CO_2_ previously absorbed by the catalyst. To this end, the catalyst was heated to 500 °C at a heating rate of 10 K min^−1^ in inert gas. At this temperature, the catalyst was reduced with 100 mL min^−1^ of H_2_ mixture for 1 h. Subsequently, the catalyst was cooled down to 50 °C under Ar. When this temperature is reached, 60 mL mi of CO_2_ was flushed for 1 h. Then, the gas was switched to 60 mL min^−1^ of 5% H_2_ in Ar and kept until the signal of the mass spectrometer was stable. Subsequently, the temperature was increased to 500 °C at a rate of 10 °C per minute while monitoring the desorbed gases, mainly CH_4_.

The structural properties of the catalyst were determined by X-ray diffraction (XRD). The XRD profiles were obtained in Polycristal X’Pert Pro PANalytical diffractometer using Ni-filtered Cu Kα radiation (λ = 1.54 Å, 45 kV and 40 mA) with a 0.04° step, and the PANalytical X´Pert HigthScore Plus software was used for the phase identification.

The X-ray photoelectron spectra of the samples after the reaction were recorded using an Omicron spectrometer refurbished by SPECS, equipped with a PHOIBOS 100 R4 analyzer and a monochromatic X-ray source (Mg Kα) operated at 75 W, with a pass energy of 30 eV and an energy step of 0.050 eV. Each sample was pressed into a small pellet, placed in the sample holder and degassed in the chamber for 6–8 h to achieve a dynamic vacuum below 10^−8^ Pa before analysis. The spectral data for each sample were analyzed using CASA XPS software. The binding energy is referenced as the Al 2p line at 74.7 eV. The relative concentrations and atomic ratios were determined from the integrated intensities of photoelectron lines corrected for the corresponding atomic sensitivity factor.

The XEDS-mapping analyses were performed in STEM mode with a probe size of ∼1 nm using the Oxford INCA Energy 2000 system detector. The samples were ground until they became a powder and were suspended in an ethanol solution using an ultrasonic bath. Then, some drops were added into the copper grid with carbon-coated layers (Aname, Lacey carbon 200 mesh), leaving to dry at room temperature to evaporate ethanol before placing in the microscope.

## 3. Results and Discussion

### 3.1. Catalytic Performance at Steady State

The mono and bimetallic catalysts were tested for isothermal CO_2_ hydrogenation feeding a gas composed of CO_2_ and H_2_ in Ar in a ratio 1:4:7 (Figure 1). The two main products detected were CO and CH_4_, which closed the carbon balance. Comparing the two monometallic catalysts, the CO_2_ conversion for 3%Ru/Al is higher than that for 15%Ni/Al despite the five-fold higher loading of the latter. For both monometallic catalysts, the CO yield is below 5% (Figure 1c,f). For the Ru-containing catalyst series, none of the bimetallic catalysts provided higher conversion than the monometallic one, while for the Ni catalyst series, several bimetallic catalysts (K and Ba-containing) exhibited higher conversions at low temperatures than the monometallic Ni catalysts. However, in terms of CH_4_ yield, only the Ba-containing catalyst outperformed the monometallic one (Figure 1b). The suppression of CO_2_ conversion and CH_4_ selectivity when adding the alkaline agrees with some of the literature [14,33]. This is usually attributed to the masking of the transition metal by the alkaline. Nevertheless, the addition of Ba to 15%Ni/Al improves the conversion while keeping selectivity to CH_4_ higher than 98%. It is important to note that while the weight loading of alkaline is the same for all materials, the atomic percentage is significantly lower for the Ba-containing one. The addition of K and Na to 15%Ni/Al, besides diminishing the conversion, also increases the yield of CO, more pronouncedly for K than for Na. This is aligned with the literature, which reports that the addition of K [12] or Na [42] to a Ni catalyst increases the selectivity to CO. The two monometallic catalysts showed selectivities to CO lower than 2%. For the Ru catalyst series, the presence of the alkaline also increases the selectivity to CO, with the lowest increase observed for the Ba-containing bimetallic catalysts. The selectivity to CO (Figure 1c,f) is higher for the Ru-based catalyst than for the Ni counterparts and increases in the same order with respect to the alkaline metal for both the Ni and Ru series, namely, no alkaline < Ba < Na < K. The selectivity is governed by the CO adsorption energy [43]. If CO adsorption is weak, the main product is CO, and if CO adsorption is strong, the main product is CH_4_. Intermediate adsorption strengths lead to mixed selectivities. It could be argued that the addition of K and Na weakens the CO adsorption strength.

The apparent activation energies for all catalysts were calculated from the Arrhenius plots (Figure 2). The apparent activation energies of monometallic Ni and Ru catalysts were 85 and 64 kJ/mol, respectively. The lower value of activation energy for Ru agrees with a lower overall reaction barrier. Lower values of Ea for Ru were also previously found by us using catalysts supported on Al_2_O_3_ washcoated monoliths, although the absolute values were ca. 10 kJ/mol smaller than here for both Ru and Ni catalysts [7]. This could be attributed to some diffusion effect for the monoliths. The values found here are in the same range as values in the literature for other Ru/Al_2_O_3_ and Ni/Al_2_O_3_ catalysts [44,45,46], although the Ea for the Ni catalyst was slightly higher here. For the bimetallic catalysts of the Ru series (Figure 2b), the Ea increases for all the alkaline metal additions, suggesting a change towards a mechanism with a higher thermodynamic reaction barrier. The lowest increase of the reaction barrier corresponds to the Ba-containing catalyst. For the Ni-series catalysts (Figure 2a), the Ba-containing catalyst keeps the same Ea as the monometallic one, suggesting that there is no change in the mechanism. However, K or Na addition have different effects on Ea. It increases upon K addition alike for the Ru-series counterpart, while Ea decreases upon Na addition in contrast to the Ru-series counterpart. This lower transition state barrier is not translated into a higher reaction rate, possibly because Na masks part of the Ni.

To compare the order of impregnation of metals, we prepared a catalyst with the same composition as (3%Ru)10%Ba/Al but using the inverse order of impregnation, i.e., first Ru and second Ba, which is the same order used to prepare the Ni-series catalysts. This catalyst, denoted as (10%Ba)3%Ru/Al, showed slower CO_2_ conversion kinetics than (3%Ru)10%Ba/Al (Figure 3a). This can probably be explained because the metal impregnated in the second step (Ba) partially masks Ru since the weight content is 3-times lower for the later metal. Therefore, for the Ru catalyst series, the methodology of adding first the alkaline and subsequently the Ru was preferred. Furthermore, no significant changes in the selectivity to CH_4_ and CO were detected (Figure 3b).

Since the catalyst (10%Ba)15%Ni/Al was the only one that outperformed its monometallic counterpart in terms of CH_4_ yield (Figure 1b), we decided to study the effect of varying the Ba loading in this catalyst. To this end, catalysts containing other Ba loadings, namely 5 wt%, 15 wt% and 20 wt%, were prepared and tested in CO_2_ hydrogenation (Figure 4). The catalyst containing 5 wt% and 10 wt% Ba showed similar conversion, and the catalyst containing 15 wt% Ba provided a significantly higher conversion. A further increase to 20 wt% did not improve the performance. The selectivity to CH_4_ in the temperature range corresponding to the highest conversions is close to 100% for all the catalysts, irrespective of the Ba loading (Figure 4b). The CH_4_ selectivity diminishes slightly for temperatures <325 °C and >475 °C for all the catalysts. At low temperatures and low CO_2_ conversion, minor amounts of undetectable products can be produced, such as CO or formic acid. At the highest temperatures, the reverse water gas shift reaction is thermodynamically favored, leading to the increase of CO selectivity. It is reported [33] that a low amount of alkaline metal is beneficial for methanation because it increments the basicity of the support. However, if the metals are present in higher amounts, they are reported to hamper methanation activity due both to an electronic effect and to a masking effect of the metal sites by the basic promoter [24,25]. In our Ni catalyst, it is apparent that the promotion effect prevails over the other detrimental effects when increasing the Ba content up to 15 wt%. A further increase to 20 wt% does not provide enhanced activity.

The enhanced performance of our Ni/Al_2_O_3_ catalyst upon Ba addition agrees with the results found with similar catalysts in the literature [36,37]. There, the increased CH_4_ yield is ascribed to the promotion of the formation of *CO and H_2_CO* intermediates and to the generation of oxygen vacancies [36]. In these works, the catalyst was prepared by impregnating first the alkaline earth metal and subsequently the nickel nitrate, in contrast to our work. One of these works reports that the catalyst deactivates due to the sintering of the nickel active phase [37]. This can be due to the fact that some nickel is deposited over the alkaline earth metal. In our catalyst, the nickel precursor is impregnated first, and it has a good interaction with the Al_2_O_3_ support, stabilizing very small nickel nanoparticles during the reaction, as observed in the characterization of the materials shown below. 

### 3.2. Basicity Assessed by Temperature-Programed Desorption of CO_2_ (CO_2_-TPD)

CO_2_ is a mildly acidic molecule, which was adsorbed on some basic sites of the catalyst under reaction conditions. The adsorbed amount also depends on the reaction temperature [4,5,6,7]. For the monometallic catalyst, CO_2_ can be adsorbed on Lewis basic sites of the metal (Ru, Ni) and on Brønsted basic hydroxyl groups of Al_2_O_3_ support [47]. Alkaline metal addition introduces supplementary absorption sites by forming metal bicarbonates and bidentate carbonates (Na_2_CO_3,_ K_2_CO_3_) [48]. To assess the amount of CO_2_ chemisorbed at temperatures kinetically relevant for methanation, we performed chemisorption of CO_2_ at 300 °C, cooling down to room temperature and subsequent TPD up to 500 °C in an Ar flow (Figure 5). The CO_2_ desorption profile shows two distinct peaks; one at a low temperature around 110–150 °C and a second at a high temperature around 350 °C. The first corresponds to physisorbed CO_2_ or bicarbonate species adsorbed on the Al_2_O_3_ support [41] (weak basic sites), while the second is chemisorbed CO_2_ (medium and strong basic sites). At the maximum temperature of TPD (500 °C), the CO_2_ concentration reaches almost zero, indicating that alkaline carbonates have been almost completely decomposed at temperatures below 500 °C. This suggests the absence of alkaline agglomerates, which decompose at higher temperatures [14,49]. The quantification of CO_2_ desorption is displayed in Figure 5c. This has been carried out by integrating the area of each peak, splitting the two peaks by a vertical line at the valley. The amount of total desorbed CO_2_ follows the same increasing trend as a function of the base metal for both Ni and Ru catalyst series, i.e., no alkaline < Ba < Na < K. The presence of the alkaline metal boosts the amount of both weak and stronger basic sites. The bimetallic catalysts of the Ru series chemisorb slightly higher amounts of CO_2_ than their counterparts of the Ni series, despite the much higher loading of the latter. This indicates that the methanation metal type also plays a role in activating/dissociating CO_2_ for subsequent chemisorption on basic sites. Surprisingly, the catalysts exhibited an inverse relationship between the amount of chemisorbed CO_2_ and the CO_2_ conversion at steady-state (Figure 1d). Therefore, the amount of chemisorbed CO_2_ is not a good descriptor for the catalyst activity at steady-state, probably because CO_2_ is adsorbed too strongly on the basic sites at reaction temperature, and thus, CO_2_ capture prevails over its conversion. Accordingly, the CO_2_ captured by K and Na forms very stable carbonates that do not contribute to the supply of CO_2_ for reaction with H_2_ at steady-state and only CO_2_ weakly adsorbed from the feed gas would be able to react. 

### 3.3. Temperature-Programed Surface Reaction of Adsorbed CO_2_ (TPSR)

The reactivity of weakly adsorbed CO_2_ for the different catalysts was assessed by TPSR (Figure 6). CO_2_ previously adsorbed on the catalyst at 50 °C is allowed to react with a H_2_ flowing gas while heating up to 500 °C at a rate of 10 °C/min. Simultaneously, the produced CH_4_ is monitored in the flue gases. The temperature of the CH_4_ peak is an indicator of the activity of the catalysts towards the hydrogenation of the CO_x_-adsorbed species. The temperature of the CH_4_ peak is about 20 °C lower and the intensity larger for the Ni-series catalysts than for their Ru counterparts. This contrasts with the higher conversion at steady-state (Figure 1) and lower activation energy of the Ru catalyst series. This could be attributed to the slightly higher reactivity of CO_2_ stored at a low temperature (50 °C) on Ni than on Ru catalysts, i.e., more stable and a higher amount of CO_2_-adsorbed species are formed on Ru catalysts than on Ni ones when adsorbed at 50 °C. What is more remarkable, the CH_4_ peak temperature was 100 °C lower for the catalyst containing Ba than for those containing the other alkaline metals for both of the Ru and Ni catalyst series. This indicates that the CO_x_-species adsorbed on Ba-containing catalysts are more reactive towards hydrogenation than those adsorbed on the other alkaline bimetallic catalysts. This is aligned with the superior performance of the Ba-containing catalysts in our steady-state experiments (Figure 1). This is at odds with other authors who observed that Ba formed more stable carbonates than Na in TPSR experiments with Ru and different alkaline [14]. This discrepancy can be explained because there are some significant differences between our experimental methodology and that of those authors. Concerning catalyst preparation, these authors deposited the alkaline after Ru and, concerning TPSR methodology; they performed CO_2_ adsorption at 500 °C instead of 50 °C in TPSR. Thus, the higher temperature can build up more stable barium carbonates. Since a reaction temperature much lower than 500 °C is preferred to perform the reaction, these very stable Ba carbonates are unlikely to be formed under operation conditions.

The CH_4_ peaks for the monometallic catalysts are very weak (left *Y*-axis in Figure 6a,b), exhibiting one order of magnitude lower intensity than the bimetallic catalyst (right *Y*-axis in Figure 6a,b). This indicates that the methanation metal (Ru or Ni) adsorbs a small amount of CO_2_ and the addition of the alkaline metal significantly enhances the adsorption of CO_2_ even at the low temperature (50 °C) used in TPSR experiments. The monometallic catalysts show two distinct CH_4_ peaks, one at high temperature similar to K and Na-containing catalysts and another at lower temperatures, namely ~250 °C for 3%Ru/Al and at 200 °C for 15%Ni/Al (weak shoulder). These low-temperature CH_4_ peaks are very close to the peak observed for the Ba-containing catalyst but are much less intense due to the lower amount of CO_2_ adsorbed. As the amount of Ba increases up to 15%, the reactivity of COx-adsorbed species increases further, as pointed out by the decrease of the CH_4_ peak temperature (Figure 6c). This explains the enhanced performance at steady-state as the Ba loading increases up to 15 wt% and rules out the formation of Ba aggregates at these loadings [14,49]. In addition, the preparation by adding first Ba and Ru second provides a higher amount of and also more-reactive Cox ad-species than the reverse order of preparation (Figure 6d), which would explain the behavior in steady-state experiments (Figure 3).

### 3.4. Catalysts Characterization by Instrumental Techniques

The mean metal particle size of the monometallic materials was 5 nm and less than 1 nm for 15%Ni/Al_2_O_3_ and 3%Ru/Al_2_O_3_, respectively (Appendix A). The materials before reaction were characterized by XEDS-mapping in STEM mode. Figure 7 shows the XEDS-mapping for the two Ba-containing materials. Representative mapping images for the other prepared materials are shown in Appendix A. The images show that both the methanation metal catalyst and the alkaline metal are homogeneously distributed throughout all the alumina surfaces, revealing an intimate contact at the nanometer scale between the two functionalities. Obviously, the metal density is higher for Ni (Figure 7a) than for Ru (Figure 7b), in agreement with the higher loading of the former.

In the XRD diffractograms of as-prepared materials (Figure 8 and Appendix A), the more intense diffraction peaks correspond to the (400) and (440) phases of γ-alumina (PDF 04-0858). The XRD pattern shows the formation of Ru nanoparticles as indicated by Ru (101) and (103) planes at 2θ values of 44.0° and 78.1° (PDF 70-0274). The presence of reduced Ru agrees with the preparation procedure, which includes a final step of the reduction in H_2_ at 500 °C. Ru can be oxidized upon exposition to air, but RuO_2_ with the main diffraction peak at 28.0° (110) (PDF 431027) is not visible in the diffractograms, suggesting that RuO_2_ is amorphous or a very small layer. For (3%Ru)10%Ba/Al, the sharp diffraction peaks at 23.8° and 42° match the (111) and (221) crystal phases of barium carbonate (PDF 45-1471) and the peaks at 23.9° and 3° can also be assigned to (111) and (211) crystal phases of barium oxides (PDF 03-0306) (Appendix A). For (3%Ru)10%K/Al, the characteristic peaks of K_2_O and K_2_CO_3_ were also observed at 28.8° (PDF 26-1327) and 31.6° (PDF 16-0820), respectively. For (3%Ru)10%Na/Al, the sharp peaks at 32° can be ascribed to Na_2_O (PDF 65-2978) [50,51].

The XRD diffractograms for the Ni-based materials (Figure 8b and Appendix A), besides the diffraction peaks corresponding to γ-alumina, show peaks at 2θ = 44.5°, 51.8° and 76.4°, corresponding to the (111), (200) and (220) crystal faces of Ni (PDF 04-0850) [52,53]. In addition, small peaks can be observed at 2θ = 37.2°, 43.3°, 62.9°, 75.4° and 79.4°, corresponding to the (111), (200), (220), (311) and (222) crystal faces of NiO (PDF 01-1239) [53], respectively. Peaks attributed to the stronger interaction between Ni and alumina, such as NiAl_2_O_4_ (PDF 01-1299), are also visible. Regarding the alkaline peaks, (10%Na)15%Ni/Al shows sharp peaks at 18.2°, 20.5° and 32°, like its Ru-series counterpart, which are ascribed to Na_2_O (PDF 23-0528). However, the peaks of K and Ba are weaker and broader than for their counterparts of the Ru series, suggesting its amorphous nature. This could be due to the fact that the alkaline was impregnated in the second step and it was only calcined but not reduced in H_2_ for materials of the Ni series.

Figure 9a shows the Ru 3d core-level spectra of the as-prepared materials reduced at 500 °C in H_2_. The K-containing material also displays a peak at 294 eV, which corresponds to the K 2p core level. The Ru 3d_5/2_ peaks at 280.3 and 280.9 eV are generally attributed to Ru^0^ [54] and RuO_2_ [55], respectively. A peak of Ru 3d_5/2_ spectrum at 282.2 eV is assigned to Ru in an oxidation state higher than IV^+^, such as Ru oxyhydroxide [56]. Above 283 eV, it is difficult to assign the Ru 3d peak because of the overlap with the C 1s peak at 284.6 eV. This carbon must come from carbonates formed spontaneously when the ex-situ reduced catalyst is exposed to the atmospheric CO_2_ [57]. Therefore, we have focused our attention on the contributions below 283 eV. The position of the peak corresponding to the more reduced species (either Ru^0^ or RuO_2_) along with the Ru/Al ratios are listed in Table 1. The monometallic Ru/Al exhibits a higher relative contribution of reduced Ru^0^ than the alkaline-containing materials. Thus, the alkaline favors the oxidation of Ru, which is deposited in the second step. Among the bimetallic materials, the Ba-containing one exhibited the lowest contribution of reduced species and the appearance of a shoulder at higher binding energies, suggesting an electron-withdrawing effect of Ba over Ru.

Regarding the Ru/Al ratio determined by XPS (Table 1), the absolute values need to be taken with caution because XPS probes only the outermost surface Ru species. However, some conclusions can be derived from relative variations. Compared to the monometallic 3%Ru/Al catalyst, the Ru/Al ratio decreases in the reduced bimetallic catalysts, which can be due to a certain Ru masking by the alkaline metal. 

Figure 9b shows the Ni 2p_3/2_ peaks, which are in the range of 851–868 eV. This region could be deconvoluted into three peaks: metallic Ni appears at 852.8 eV [58], NiO at 853.3 eV [59,60], and Ni_2_O_3_ and Ni(OH)_2_ are found at about the same value of 856.3 eV [53,54,55]. Moreover, shake-up satellites can be seen at a higher binding energy of 6.1 eV. In our materials, the presence of the most reduced Ni species (Ni^0^ or NiO) is minor. Therefore, Ni is reoxidized after air exposure, even more after the addition of the alkaline in the second step. The maximum of the Ni peak at 856–859 eV indicates that the major part of Ni is present as Ni_2_O_3_ or Ni(OH)_2_. Comparing the bimetallic materials with the monometallic 15%Ni/Al, the Ba and K-containing materials exhibited a shift of the maximum towards more oxidized species, suggesting an electron-withdrawing effect of Ba over Ru. In contrast, Na apparently exerts a slight electron-donating effect.

All the characterization results lead us to propose a mechanism (Figure 10). First, CO_2_ is adsorbed to form carbonated alkaline species. Subsequently, these species decompose to provide activated COx species to the dissociative adsorbed H_2_ on the metal Ni or Ru, where CH_4_ or CO are formed. In the case of Ba, the carbonated species are decomposed at lower temperatures in an H_2_ atmosphere (route A in Figure 10) than for the other alkali metals (route B in Figure 10), explaining enhanced kinetics at steady-state.

## 4. Conclusions

Bimetallic catalysts based on a combination of a methanation catalyst (Ni or Ru) and an alkali (Na or K) or alkaline earth (Ba) metal supported on alumina have been scrutinized for the Sabatier reaction. For the Ni-based catalyst series, a 10 wt% of alkaline metal was added in a second step to 15 wt% Ni on Al_2_O_3_. In the case of bimetallic catalysts containing a 3 wt% Ru on γ-alumina, higher activity was attained by the reverse order of impregnation, first adding the alkaline metal and second the Ru precursor. The characterization of the materials showed a homogenous dispersion of the two metals over the alumina support, guaranteeing a nanometer-scale interaction between both metals. The addition of the base metal boosted the CO_2_ chemisorption at temperatures relevant for CO_2_ methanation (300 °C). However, this enhanced CO_2_ chemisorption did not always entail a higher CO_2_ conversion for the Ru catalyst series in a methanation reaction. In fact, the Ba-containing catalyst chemisorbed the least amount of CO_2_ among the bimetallic catalysts but provided the best performance. Comparing the Ni catalyst with and without alkaline, the addition of Na or K improved the kinetics in the low-temperature region, but the selectivity to CO increases compared to the monometallic catalyst. Only the addition of Ba to the 15%Ni/Al_2_O_3_ catalyst enhanced the CO_2_ conversion in the whole temperature range with respect to the monometallic catalyst keeping almost 100% selectivity to CH_4_. According to TPSR characterization, the Ba-containing catalyst affords higher methanation kinetics of COx ad-species than those for the K or Na-containing catalysts. This suggests that the enhanced performance in CO_2_ methanation at steady-state is a trade-off between a higher amount of chemisorbed CO_x_ and higher reactivity of the COx ad-species. The optimum of the two parameters is met here for barium-containing catalysts. In future work, we will explore the use of these bifunctional materials for stepwise CO_2_ capture and subsequent methanation of captured CO_2_.

## Figures and Tables

**Figure 1 nanomaterials-12-01052-f001:**
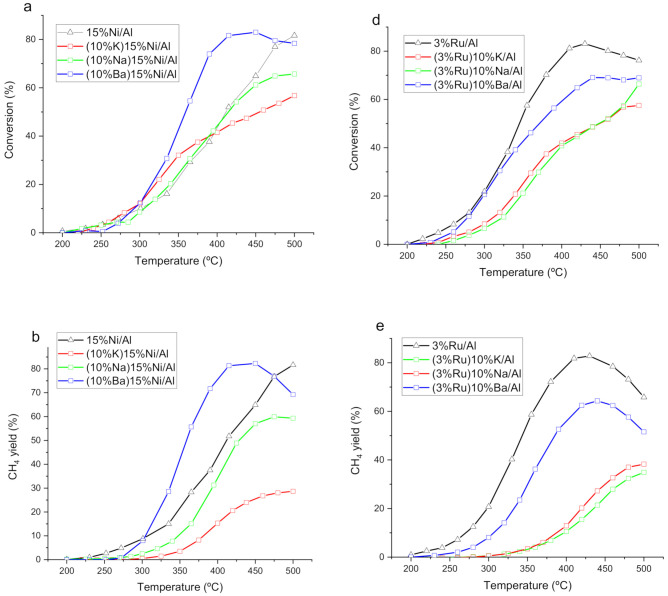
CO_2_ conversion, CH_4_ and CO yield for catalyst based on Ni (**a**–**c**) and Ru (**d**–**f**) without alkaline metal and with three alkaline metals (K, Na, Ba).

**Figure 2 nanomaterials-12-01052-f002:**
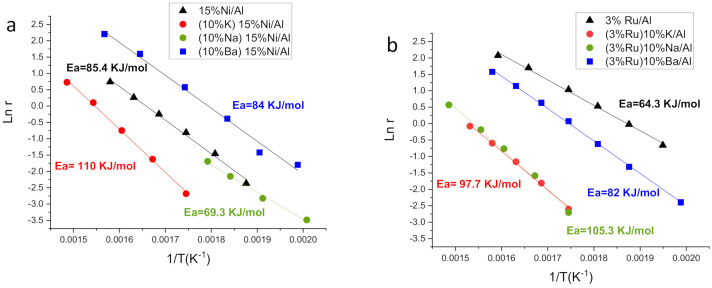
Arrhenius plot with apparent activation energies for catalysts of: (**a**) Ni series and (**b**) Ru series.

**Figure 3 nanomaterials-12-01052-f003:**
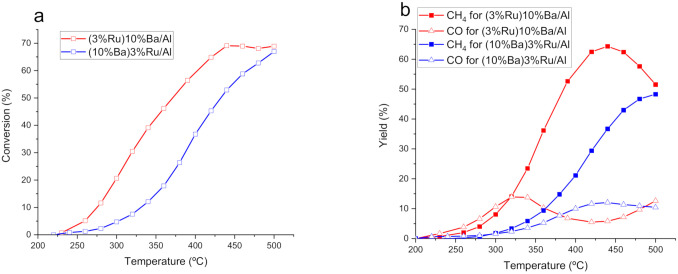
Comparison of the performance of the order of addition of the methanation metal and alkaline metal: (**a**) conversion, (**b**) CH_4_ (filled symbols) and CO (empty symbols) yield.

**Figure 4 nanomaterials-12-01052-f004:**
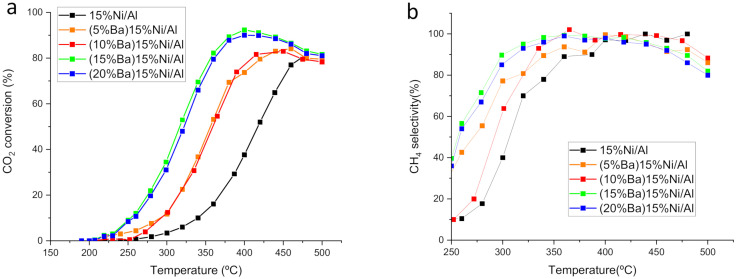
Effect of the amount of Ba added to 15%Ni/Al: (**a**) CO_2_ conversion; (**b**) Selectivity to CH_4_.

**Figure 5 nanomaterials-12-01052-f005:**
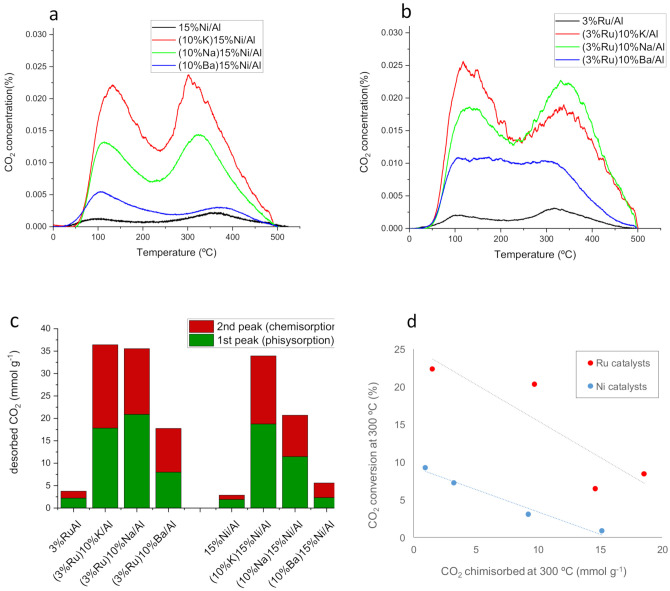
CO_2_-TPD profiles for Ni (**a**) and Ru (**b**) series catalysts; (**c**) quantification of CO_2_ physisorbed and chemisorbed; (**d**) steady-state CO_2_ conversion at 300 °C as a function of chemisorbed CO_2_.

**Figure 6 nanomaterials-12-01052-f006:**
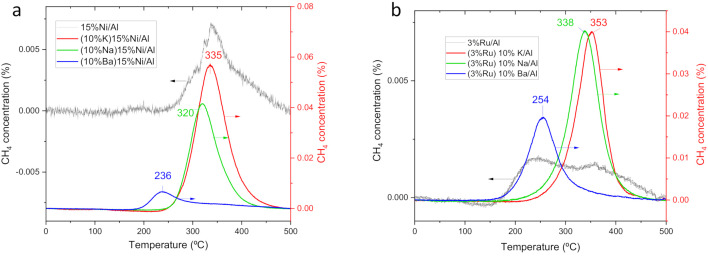
Temperature-programed surface reaction (TPSR) experiments of CO_2_ pre-adsorbed at 50 °C on the different catalysts. (**a**) Comparison of catalysts of the Ni series, (**b**) comparison of catalysts of the Ru series, (**c**) comparison of catalysts containing different amounts of Ba, (**d**) comparison of Ba-Ru catalyst prepared changing the order of impregnation of the two metals.

**Figure 7 nanomaterials-12-01052-f007:**
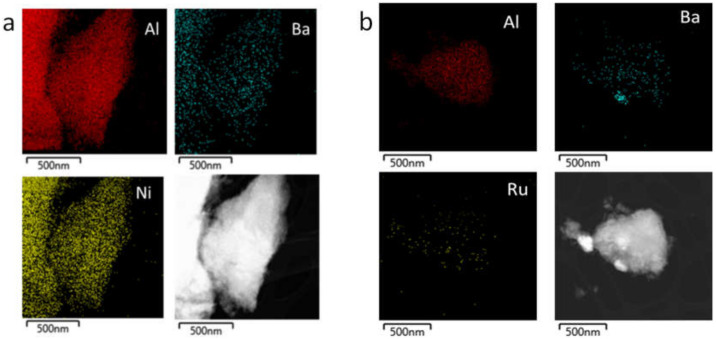
Representative XEDS-mapping in STEM mode for: (**a**) (10%Ba)15%Ni/Al and (**b**) (3%Ru)10%Ba/Al.

**Figure 8 nanomaterials-12-01052-f008:**
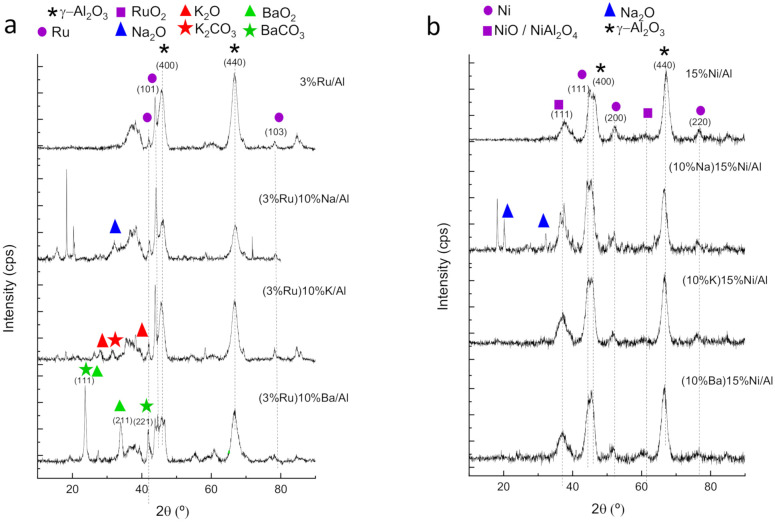
X-ray diffraction peaks for monometallic and bimetallic catalysts before use: (**a**) Ru-based materials, (**b**) Ni-based materials.

**Figure 9 nanomaterials-12-01052-f009:**
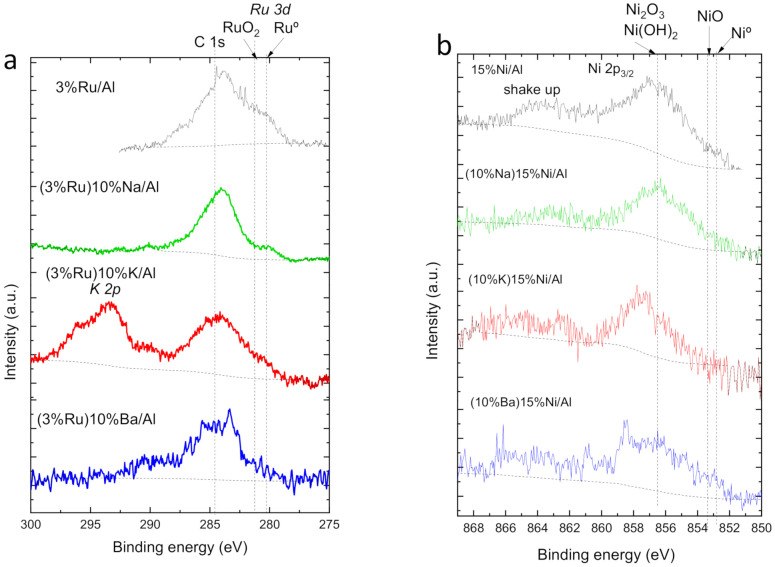
X-ray photoelectron spectroscopy characterization of the as-prepared materials before reaction: (**a**) Ru 3d core level of Ru-based materials, (**b**) Ni 2p_3/2_ of Ni-based materials.

**Figure 10 nanomaterials-12-01052-f010:**
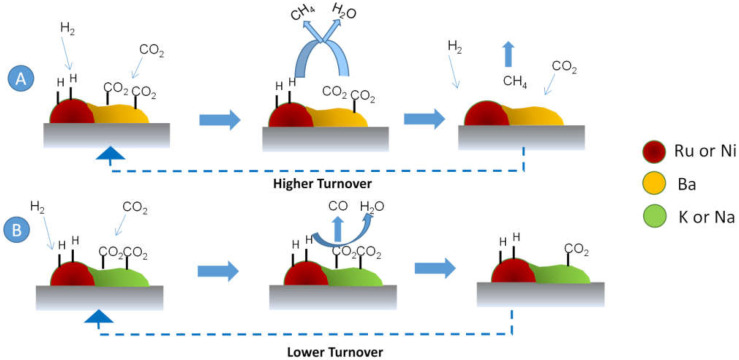
Schematic representation of the reaction mechanism.

**Table 1 nanomaterials-12-01052-t001:** XPS data of reduced catalyst previously to reaction.

	Peak Maximum Position	M/Al (at)	Al/O (at)
3%Ru/Al	281.2	0.119	0.56
(3%Ru)10%Ba/Al	282.1	0.046	0.51
(3%Ru)10%K/Al	281.0	0.035	0.48
(3%Ru)10%Na/Al	280.3	0.068	0.37
15%Ni/Al	856.8	0.061	0.57
(10%Ba)15%Ni/Al	858.5	0.084	0.39
(10%K)15%Ni/Al	857.7	0.025	0.55
(10%Na)15%Ni/Al	856.3	0.038	0.48

M = Ru or Ni.

## Data Availability

Not applicable.

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
