# Peer review of "Promotion of Ru or Ni on Alumina Catalysts with a Basic Metal for CO2 Hydrogenation: Effect of the Type of Metal (Na, K, Ba)"

_nanomaterials, 2022, doi:10.3390/nano12071052_

Round 1
Reviewer 1 Report
Dear E. García-Bordejé;
On behalf of your submission to MDPI Nanomaterials with control ID: nanomaterials-1652287. Here are some comments in order to improve the quality of your submission.
You paper, presents chemical synthesis and catalytic test of promoted systems, however:
1) Improve material characterization, mainly if possible to perform some TEM (if the instrument not available, totally understand).
2) Would be best if the "spent" catalyst is also observed, to detect is any "coke" formation deactivates the material.
3) It seems that Ruthenium its better catalytic material, something that has been published extensively by others (Chianelli et al. "Sabatier Principle"), however on conclusion is a bit contradictory when indicating that Ni has better selectivity. Please clarify the statements.
4) Use an appropiate indexing on all chemical formulas, on the references subindex aren't place accordingly.
Author Response
We thank the reviewers for the comments that have help to improve the manuscript. The changes are highlighted in red in the revised manuscript and the changes to the specific comments are detailed below.
Reviewer 1.
You paper, presents chemical synthesis and catalytic test of promoted systems, however:
- Improve material characterization, mainly if possible to perform some TEM (if the instrument not available, totally understand).
We can not perform TEM currently of all samples because we have run out of samples in some cases. We have performed TEM of the monometallic catalyst and it is included in supplementary information: Figures S1 y S2.
- Would be best if the "spent" catalyst is also observed, to detect is any "coke" formation deactivates the material.
The spent catalyst was studied by XPS. The main deactivation is caused not by coke deposition because the base metal does not favour reactions leading to coke. The main deactivation is caused by carbonate formation as observed by XPS
- It seems that Ruthenium its better catalytic material, something that has been published extensively by others (Chianelli et al. "Sabatier Principle"), however on conclusion is a bit contradictory when indicating that Ni has better selectivity. Please clarify the statements.
The sentence in the conclusions does not says that Ni has better selectivity than Ru but says that the Ni withput the alkaline has better selectivity than with the alkaline. In case that the sentence is not clear enough it has been modified. Now it says: “Comparing the Ni catalyst with and without alkaline, the addition of Na or K improved the kinetics in the low temperature region but the selectivity to CO increases compared to the monometallic catalyst”
4) Use an appropiate indexing on all chemical formulas, on the references subindex aren't place accordingly.
Thank you for the comments. The subidex in references have been corrected.

Reviewer 2 Report
The paper titled “Promotion of Ru or Ni on alumina catalysts with a basic metal for CO2 hydrogenation: effect of the type of metal (Na, K, Ba)” is aimed to reveal the real situation for the case of Ru/Al2O3 and Ni/Al2O3 promotion with alkali and alkaline earth metals, while the literature data are inconsistent. The effects of the order of the metal deposition via the impregnation technique and the promoter loading were investigated within this research. The paper is well written and presents new experimental data on the CO2 conversion into methane. Therefore, I recommend this paper for publication after major revision. The comments are listed below.
- Page 2, line 19: Check the lower index in CO2.
- Page 2, lines 20-21: “It was found that the performance of the monometallic catalyst in CO2 hydrogenation decreased …” What is the decreased performance? May be worsened?
- “Takano et al. found and optimum Ca concentration” -> “Takano et al. found an optimum Ca concentration”
- Some sentences look strange and confused in meaning. The modification of some catalyst with an additive by the impregnation procedure means that the catalyst is impregnated with the solution of this additive. Therefore, for instance, the following sentence “Cimino et al.34 impregnated different alkaline metals (Li, Na, K) over 1%Ru/Al2O3 and only found a promotion when Li was impregnated” should be “Cimino et al.34 impregnated 1%Ru/Al2O3 with different alkaline metals (Li, Na, K) and found a promotion only when Li was added”
- “Thus, further research is needed about the promotion with heavier alkaline earth metals (Ba, Cs) and about their comparison with the promotion by alkali metals.” – Cs is not mentioned neither in the literature overview nor in the present research.
- Page 3: “was calcined at 500 ºC under N2 for 1 hour” -> “was calcined at 500 ºC under N2 for 1 h”
- The last paragraph of the Experimental part seems to be incomplete.
- More information (manufacturer, city, country) should be given for all the apparatuses mentioned in the Experimental part.
- The loading of promoter was varied. This should be pointed out in the abstract. Now, it is stated that Ru and Ni on alumina catalysts have been promoted with a 10 wt% of alkali metal (K or Na) or alkaline earth metal (Ba).
- Figure 4b: The legend is missing.
- Figure 5c shows data for the (10%M)3%Ni/Al. Is it not a misprint? No information is given about the 3%Ni/Al catalyst.
- Figure 5c: There is also some confusion with the colors of the bars. Green means the first peak, and red means the second one. This does not correspond to the peak intensities in Figure 5a.
- Scheme in Figure 10: When CO2 goes out as CO, the path for the released oxygen should be shown.
- What is about water in the gas-phase products?
- “If CO adsorption is weak the main product is CO and if CO adsorption is strong, the main product is CH4.” – missing commas -> “If CO adsorption is weak, the main product is CO, and if CO adsorption is strong, the main product is CH4.”
- All over the text: “KJ/mol” -> “kJ·mol-1”
- “Ru can oxidise upon re-exposition” -> “Ru can be oxidized upon re-exposition”
Author Response
We thank the reviewers for the comments that have help to improve the manuscript. The changes are highlighted in red in the revised manuscript and the changes to the specific comments are detailed below.
Reviewer 2
The paper titled “Promotion of Ru or Ni on alumina catalysts with a basic metal for CO2 hydrogenation: effect of the type of metal (Na, K, Ba)” is aimed to reveal the real situation for the case of Ru/Al2O3 and Ni/Al2O3 promotion with alkali and alkaline earth metals, while the literature data are inconsistent. The effects of the order of the metal deposition via the impregnation technique and the promoter loading were investigated within this research. The paper is well written and presents new experimental data on the CO2 conversion into methane. Therefore, I recommend this paper for publication after major revision. The comments are listed below.
- Page 2, line 19: Check the lower index in CO2.
- Page 2, lines 20-21: “It was found that the performance of the monometallic catalyst in CO2 hydrogenation decreased …” What is the decreased performance? May be worsened?
- “Takano et al. found and optimum Ca concentration” -> “Takano et al. found an optimum Ca concentration”
- Some sentences look strange and confused in meaning. The modification of some catalyst with an additive by the impregnation procedure means that the catalyst is impregnated with the solution of this additive. Therefore, for instance, the following sentence “Cimino et al.34 impregnated different alkaline metals (Li, Na, K) over 1%Ru/Al2O3 and only found a promotion when Li was impregnated” should be “Cimino et al.34 impregnated 1%Ru/Al2O3 with different alkaline metals (Li, Na, K) and found a promotion only when Li was added”
- “Thus, further research is needed about the promotion with heavier alkaline earth metals (Ba, Cs) and about their comparison with the promotion by alkali metals.” – Cs is not mentioned neither in the literature overview nor in the present research.
- Page 3: “was calcined at 500 ºC under N2 for 1 hour” -> “was calcined at 500 ºC under N2 for 1 h”
- The last paragraph of the Experimental part seems to be incomplete.
- More information (manufacturer, city, country) should be given for all the apparatuses mentioned in the Experimental part.
We thank the reviewer for pointing these mistakes in points 1 to 8, which have been corrected in the revised manuscript.
- The loading of promoter was varied. This should be pointed out in the abstract. Now, it is stated that Ru and Ni on alumina catalysts have been promoted with a 10 wt% of alkali metal (K or Na) or alkaline earth metal (Ba).
Thank you for this comment. The following sentence has been introduced in the abstract:
“For the catalyst consisting of Ni and Ba, the Ba loading was varied keeping Ni loading constant.”
- Figure 4b: The legend is missing.
The legend has been added
- Figure 5c shows data for the (10%M)3%Ni/Al. Is it not a misprint? No information is given about the 3%Ni/Al catalyst.
Thank your for noticing this misprint. It has been corrected
- Figure 5c: There is also some confusion with the colors of the bars. Green means the first peak, and red means the second one. This does not correspond to the peak intensities in Figure 5a.
Figure 5c shows the quantification based in the areas of the peaks, not based on the intensities. We have explained in the revised manuscript how the quantification in Figure 5 b has been carried out.
- Scheme in Figure 10: When CO2 goes out as CO, the path for the released oxygen should be shown.
The path to oxygen is H2O evolution. This has been depicted in Figure 10
- What is about water in the gas-phase products?
The product water was missing in Figure 10 and it has been included in revised version. Water in the gas phase products is swept with the flow and condensed downstream
- “If CO adsorption is weak the main product is CO and if CO adsorption is strong, the main product is CH4.” – missing commas -> “If CO adsorption is weak, the main product is CO, and if CO adsorption is strong, the main product is CH4.”
Thank you the reviewer. The commas have been added
- All over the text: “KJ/mol” -> “kJ·mol-1”
This has been corrected all over the text
- “Ru can oxidise upon re-exposition” -> “Ru can be oxidized upon re-exposition”
This has been corrected
Round 2
Reviewer 2 Report
The authors have revised the manuscript thoroughly. Most of the comments are addressed. There are few corrections listed below that can be made at the proofreading stage. The paper can be accepted for publication.
- Page 3: “was calcined at 500 ºC under N2 for 1 hour” -> “was calcined at 500 ºC under N2 for 1 h”
- More information (manufacturer, city, country) should be given for all the apparatuses mentioned in the Experimental part.
- Figure 2: “KJ/mol” -> “kJ/mol”